# Neuro-Evolutionary Framework for Design Optimization of Two-Phase Transducer with Genetic Algorithms

**DOI:** 10.3390/mi14091677

**Published:** 2023-08-27

**Authors:** Aneela Zameer, Sidra Naz, Muhammad Asif Zahoor Raja, Jehanzaib Hafeez, Nasir Ali

**Affiliations:** 1Department of Computer and Information Sciences, Pakistan Institute of Engineering and Applied Sciences, Nilore, Islamabad 45650, Pakistan; 2Department of Electrical Engineering, Pakistan Institute of Engineering and Applied Sciences, Nilore, Islamabad 45650, Pakistan; sidranaz_19@pieas.edu.pk; 3Future Technology Research Center, National Yunlin University of Science and Technology, 123 University Road, Section 3, Douliou, Yunlin 64002, Taiwan

**Keywords:** piezoelectric, multilayer transducer, piezocomposite, optimization, genetic algorithm

## Abstract

Multilayer piezocomposite transducers are widely used in many applications where broad bandwidth is required for tracking and detection purposes. However, it is difficult to operate these multilayer transducers efficiently under frequencies of 100 kHz. Therefore, this work presents the modeling and optimization of a five-layer piezocomposite transducer with ten variables of nonuniform layer thicknesses and different volume fractions by exploiting the strength of the genetic algorithm (GA) with a one-dimensional model (ODM). The ODM executes matrix manipulation by resolving wave equations and produces mechanical output in the form of pressure and electrical impedance. The product of gain and bandwidth is the required function to be maximized in this multi-objective and multivariate optimization problem, which is a challenging task having ten variables. Converting it into the minimization problem, the reciprocal of the gain-bandwidth product is considered. The total thickness is adjusted to keep the central frequency at approximately 50–60 kHz. Piezocomposite transducers with three active materials, PZT5h, PZT4d, PMN-PT, and CY1301 polymer, as passive materials were designed, simulated, and statistically evaluated. The results show significant improvement in gain bandwidth compared to previous existing techniques.

## 1. Introduction

Ultrasonic transducers have been widely developed due to their versatile capabilities, high sensitivity, low impedance, and accurate investigative applications. As the core element of many exploratory systems, ultrasonic transducers have excellent properties in vast medical and engineering fields, including energy harvesting, nondestructive evaluation, underwater sonar, bone damage detection, physical acoustics, medical ultrasound technologies, son chemical and material characterization, aircraft [1,2], and many more. The development of the transducer is carried out by studying the main design parameters of the transducer, such as the matching layer material, piezoelectric stack, backing layer material, central frequency, transducer diameter, acoustic focal length, bondline thickness, bandwidth, and resolution. Among them, the piezoelectric stack material is key to the excellence of the transducers. The widely used piezoelectric materials in high-frequency transducers are generally piezoceramic lead zirconate titanate (PZT) [3] and single crystals lead magnesium niobate–lead titanate (PMN-PT) [4] due to their high performance and easy manufacturing procedure. Piezoelectric materials have a high electromechanical coupling factor, high piezoelectric voltage constants, and high dielectric constants. These materials can potentially deal with the considerably enhanced performance of ultrasonic transducers through a large bandwidth and high sensitivity for both transmission and reception configurations [5]. Ceramic PZT transducers are used for energy harvesting [6], structural health monitoring [7], damage detection in concrete for constructional applications [8], and ultrasonic diagnostic techniques in medical applications [9]. Recently, single crystal PMN-PT material-based transducers have received the most attention owing to their high attenuation, and enhanced bandwidth is used in many applications, such as underwater acoustics [10], biomedical applications, and energy harvesting [11]. These piezoceramic and single crystal materials are also combined with polymers to form piezocomposites with various volume fractions to obtain more efficient and accurate results. The combination may be a 1–3 composite or 2–2 composite depending on the dimensions of the material. These materials have superior piezoelectric responses. The excellent properties of these composite based transducers have been studied underwater ultrasound [12], medical ultrasound, nondestructive evaluation applications [13], structural safety, personalized recognition and human–machine interaction [14], and nano-positioning [15].

The optimization of transducers has been performed for specific applications to minimize the input of the transducer and enhance the target detection sensitivity and coupling coefficient constant. The nature of the optimization problem has various categories, and they may be separable, no separable, unimodal, multimodal, scalable, or hybrid. These numerical optimization problems can be solved by evolutionary algorithms.

Evolutionary algorithms are frequently used for solving multimodal, nonlinear, constrained, unconstrained, and non-differentiable intricate numerical optimization problems [16]. They are population-based, stochastic search, iterative mechanisms, looking for global minima belonging to the given problem [17]. The demonstration of many solutions for statistical optimization has been performed by a pattern matrix that contains the dimensions of the problem [18]. Evolutionary algorithms are classified into three broad categories: nature-inspired, swarm-inspired, and bioinspired [19]. Simulation of these frequently used algorithms involves selection, mutation, and crossover genetic processes; however, their success depends on the structural properties, initial population, problem dimensions, number of iterations for function evaluation, random values generator, and objective function [20]. These stochastic techniques are successively used to find optimum solutions for numerous science and engineering design problems, such as speech recognition [21], sensor development problems [22], data mining [23], video processing [24], nano/micro positioning devices [25], and many other applications. Hence, differential evolutionary approaches are statistically powerful and extensively used soft computing techniques to solve real-valued statistical optimization problems.

These methodologies have substantial applications in the field of ultrasonication. Different experiments were performed successively by using these evolutionary algorithms for the minimization of transducer input and maximization of output pressure; internal design parameters for uniform and nonuniform layer transducers, such as the optimization of active material thickness and electrical impedance; and design development with enhanced sensitivity, broadband, and electromechanical coupling coefficients [22,26]. Particle swarm optimization is used to optimize the effects of internal defects of damaged composite patches and hence improve the overall performance of transducers [27]. Simulated annealing is used for multilayer piezoelectric transducers to achieve broad bands [28]. However, a five-layer piezocomposite transducer has not been investigated yet to get optimized layer thicknesses and volume fractions to get maximum output in the form of bandwidth and sensitivity.

The GA is used in many applications, such as game theory [29], polymer design [30], multi-objective problems [31], lung cancer prognosis [32], wind power prediction [33], social networks [34], combustion engine [35], photovoltaic systems [36], task scheduling [37], traffic flow model [38], automotives [39], heat transfer [40], Complex networks [41], traffic management [42], Plasticity Echo State Network [43], prostate cancer [44], pruning for neural network [45], wireless sensor networks [46], Virtual machine [47], electric vehicles [48], IOT network topologies [49], stock market forecasting model [50], and task assignments to agents [51].

Motivated by the above perspectives, this work presents the optimization of piezocomposite transducers by varying the volume fraction of active material and layer thicknesses for highly sensitive configurations. Mathematical optimization for high mechanical output and maximum bandwidth gain was performed by a genetic algorithm for a five-layer piezocomposite transducer. The composite consists of both active and passive materials. The active materials used in this optimized design are PZT5h, PZT4d, PMN-PT, and passive material CY1301. For the first time, a 10 variable optimization design for non-uniform layer thicknesses is presented for a five-layer transducer. The first five variables are the layers’ thicknesses while the other five are the volume fraction of piezoelectric material in these layers. The most demanding transducers for advanced applications must have high sensitivity, low input voltage, and high mechanical output. For this purpose, optimization is carried out by increasing the number of layers N, which increases the capacitance of the ceramic and, hence, amplifies the output pressure by N and degrades the electrical impedance by 1/N^2^ for the same applied volts. These multiple layers are electrically connected in parallel while mechanically connected in series. The salient features of the proposed work are the following:A multilayer transducer with five layers has been selected for optimizing the wide bandwidth for underwater applications. PZT5h-, PZT4d-, and PMN-PT-based piezocomposite materials are considered active materials in each layer;Ten design variables, including five-layer thicknesses and five piezoceramic volume fractions, are considered for gain-bandwidth optimization through genetic algorithms while the ODM is used to calculate the transducer output;The transducer operating approximately 50 kHz with a wider bandwidth is optimized using GA and its five variants;Evaluation of the proposed technique is carried out with published results from simulated annealing for PZT5h and PZT4d;A novel piezocomposite is proposed to be used as an active material for better and wider bandwidth than piezoceramics.

The paper is organized as follows. Initially, the mathematical formulation of a piezocomposite non-uniform multilayer transducer by a one-dimensional model (ODM) along with the optimization strategy proposed by the GA is presented in Section 2. Proposed scenarios for the optimization of piezocomposite transducers and their simulated results are discussed in Section 3. Finally, the conclusion of the proposed work is summarized in Section 4.

## 2. Materials and Methods

In this section, the proposed methodology is presented, which comprises a system modeling overview, mathematical formulation, scope and margins of one-dimensional modeling, optimization procedure entailing formulation of an objective function, optimization steps of bioinspired heuristic genetic algorithms, and graphical abstract of proposed work is shown.

In this paper, an ultrasonic transducer containing a stack of five non-uniform piezocomposite layers was considered, which is demonstrated in Figure 1. In Piezocomposite transducer, each layer of stack consisted of 1–3 connectivity of piezoelectric ceramic-/single crystal-polymer composite materials. The piezo ceramic materials PZT5h, PZT4d, and single crystal PMN-PT were used as an active material while polymer CY1301 was used as a passive material, and characteristics of these materials are given in Table 1 and Table 2. The five layers of the stack were mechanically connected with each other in series while electrically connected in parallel. The modeling of this piezocomposite transducer was performed by a one-dimensional model. Where the layers’ thicknesses, the volume fraction of active material and its properties, front and back loading media, and electrical circuit were provided as input parameters. Then, ODM calculated the mechanical output of the transducer in transmission mode for water–air loading medium. The water was kept as a front-loading media while air was kept as back loading media. The properties of water-air loading media are given in Table 3. The study of piezocomposite transducers was investigated to discover applications in low frequency and broadband devices for underwater SONAR systems. A complete workflow diagram of the proposed methodology is shown in Figure 2.

### 2.1. Mathematical Formulation by One-Dimensional Model

Initially, a one-dimensional model of a piezocomposite transducer provided the solution of the unidimensional wave equation using matrix operations. The lateral dimensions (length and width) of the piezocomposite stack of the transducer were 10% to 20% larger than the thickness. The generated ultrasonic waves were transmitted in addition to the direction of the stack thickness in the form of longitudinal plane waves and were mathematically represented in Equations (1) and (2). These equations can be represented with variable *z* in the form of differential equations, where *z* is the thickness dimension of the transducer and are shown in Equations (3) and (4). Using Newton’s second law of motion, a one-dimensional wave equation was obtained, as given in Equations (5) and (6). The symbols used in these equations are provided in the nomenclature, where symbol *S* represents the strain vector of mechanical output, *S_V_* demonstrates the coefficient vector at zero or constant electric field, *T* symbolizes stress vector of mechanical output, *V* represents electric field, *D* indicate charge density vector, *d^t^* shows strain constants matrix at zero or constant strain, $\varsigma_{T}$ signifies relative permittivity at zero and constant strain, *h* denotes the piezoelectric constant, *c* is the elastic constant, ϖ representing the displacement of particle, *z* is the flow direction of wave propagation thru each layer of piezoelectric material stack, and ℓ (m/s) determines the velocity of particle that is intended from the significant elastic modulus *c* and the density of piezoelectric material *ρ*. Symbol *t* indicates the layer passage time of the wave.
(1)$$S=s_{V}\cdot T+d^{t}\cdot V$$
(2)$$D=d\cdot T+\varsigma_{T}\cdot V$$
(3)$$T=c\frac{\partial \varpi }{\partial z}-hD$$
(4)$$V=-h\frac{\partial \varpi }{\partial z}+\frac{D}{\varsigma_{T}}$$
(5)$$\frac{\partial^{2}\varpi }{\partial t^{2}}=\ell^{2}\frac{\partial^{2}\varpi }{\partial z^{2}}\;\mathrm{where}\;\ell^{2}=\frac{c}{\rho }$$

The ODM is a computational tool that is designed to provide the solution of the wave equation in the form of matrices. To execute time–domain analysis, the inverse Fourier transformation for a wide range frequency is performed; therefore, ODM is equivalent to the formulation of linear systems. The ODM programming comprises the following three main steps:Preprocessing: The geometrical and physical shape of the transducer, its dimensions, the number of layers of piezoceramic stack, active and passive materials for composite structures, electric circuit, numerical parameters, and front and back acoustic loading media are fed to the ODM;Processing: Matrix manipulation is performed to solve the wave equations;Post-processing: Desired mechanical outputs are stored in data files that can be graphically read.

Many researchers have performed experiments to validate ODM [52,53]. It can be used for both transmission and reception applications; as it does not consider lateral dimensions, it cannot perform resonance activities. It gives mechanical output for transducers containing simple and composite multilayer thicknesses (which may be uniform or non-uniform), active and passive layers, matching layers, bondlines, backing layers, different front and back loading media, and connections with external electrical circuits.

### 2.2. Optimization by Genetic Algorithm

The proposed methodology consists of a stochastic approach genetic algorithm (GA) [54] that is used for the optimization of thicknesses and volume fractions. The initial values of the parameters used in the genetic algorithm are numerically shown in Table 4. The steps of the proposed methodology are given as follows:

Initialization: The population of 10 variables was initialized randomly. A generated set of 10 variables consisting of 5 layered thicknesses with 5 volume fractions were then normalized according to the defined upper and lower limits. These ten variables were then given as input to the one-dimensional model (ODM) to calculate the mechanical output information of the pressure magnitude and electrical impedance magnitude. Fitness evaluation was performed in two steps that are given below.
Maximum Bandwidth: The ODM executed the temporary file that contained 10 generated variables and then computed the magnitude of pressure peaks and electrical impedance peaks for 32,768 frequency samples to obtain the maximum bandwidth. The implemented control program was designed to calculate the higher and lower frequencies (*f_h_* and *f_l_*) from the obtained maximum pressure peaks to be used to evaluate the central frequency (*f_c_*) by Equation (6);
(6)$$f_{c}=\frac{f_{h}+f_{l}}{2}$$
II.Compensation: The obtained central frequency may exceed 60 kHz. Therefore, we need to compensate for the frequency in the range of 50 kHz to 60 kHz by using Equation (7). The nonuniform five-layer thicknesses of the piezocomposite transducer were updated by taking the product with a compensated value and then used as an input to the ODM to calculate the mechanical output of the ultrasonic transducer.
(7)$$compensation=\frac{f_{c}}{50k}$$

The implemented optimization program extracted the concerned data from ODM output files and then calculated the fitness function value for the current generation by using the objective function (*O_ft_*) given in Equation (8):(8)$$O_{ft}=\frac{1}{P_{m}\times Gbw}$$
where *P_m_* was the maximum output pressure of the transducer and *Gbw* was the gain bandwidth obtained during successive generations.

Reproduction process: The genetic algorithm contained the reproduction process, which was performed in three major sub-steps as follows:Selection: A population for the next generation was produced by the recombination of chromosomes that were selected according to their finesses. The recombination process was performed by the crossover of parents and mutation within chromosomes;Crossover: The new offspring were produced by a combination of genes that were selected from the pair of current chromosomes;Mutation: A random change took place in the genes of individual chromosomes to produce new offspring and avoid trapping the solution in local minima.

Fitness evaluation: In this step, the fitness value of the newly generated population (thicknesses) was evaluated, and then, the individuals with the best fitness value were selected for the next step.

Scaling: In this step, the achieved fitness values were renewed into a suitable range of selection criteria.

Termination: The execution of the program was performed until any one of the stopping criteria was met. The stopping criteria were as follows:A maximum number of generations reached;The fitness value does not improve for 30 continuous generations;The required fitness value is achieved.

## 3. Results and Discussion

In this section, the results from three types of piezocomposite ultrasonic transducers are presented. A control program is written to run GA to generate random thicknesses and ceramic volume fractions of five layers of the transducer, which are passed to the ODM to produce mechanical output. The fitness function is then evaluated for each structure, and thickness values are adjusted according to the volume fraction as the velocity within the composite is changed accordingly and, hence, the thickness values. The first piezocomposite transducer was designed based on piezoceramic material PZT4d as the active material along with polymer (CY1301) as a passive material. The second transducer was designed based on the PZT5h material while the third transducer contained the single crystal material PMN-PT along with the polymer. The active materials were electrically connected in the circuit to obtain the input voltage. All transducers were analyzed for water–air as the loading medium, where the mechanical pressure was observed and calculated in the font loading medium, i.e., water while air is kept as a back-loading media. Each transducer contains a 5-layer piezocomposite stack. The thickness of each layer and its volume fraction of polymer were optimized by the evolutionary approach genetic algorithm (GA) for populations 20, 30, and 40. The mechanical output of the ODM is used by the implemented control program to calculate the central frequency (fc), gain bandwidth (Gbw), and fitness function values for all piezocomposite transducers based on the PZT4d, PZT5h, and PMN-PT materials, as shown in Table 5. Where the variables ‘th’ represents the optimized layer thicknesses of five layers, vf represents the volume fraction of active piezoelectric material in the corresponding layer, and fitness is the minimum objective function value achieved by optimization. These transducers are widely used for underwater SONAR. The piezocomposite transducer containing the PMN-PT shows a higher gain bandwidth and central frequency than the other two PZT4d and PZT5h based transducers. The best fitness function value for PZT4d material-based transducer is 0.001189 which is achieved in 20 populations while the PZT5h material-based piezocomposite transducer is 0.000732, which is obtained in 20 populations; however, the PMN-PT material-based piezocomposite transducer shows a higher fitness value that is 0.000680 in 30 populations.

The optimized layer thickness comparison for all three transducers is demonstrated graphically in Figure 3a, and the layer thicknesses along with their volume fraction are shown in Figure 3b. The optimized volume fraction for the corresponding layers of all three types of transducers is shown in Figure 3c, statistical evaluation of fitness values for each generation of the GA1 variant is graphically demonstrated in Figure 3d, and the pressure magnitude of the PMN-PT-based transducer in dB is shown in Figure 3e, and the central frequencies for various populations of the three proposed transducers are shown in Figure 3f.

### 3.1. Variants of GA

In this section, we consider the five different cases of GA variants according to the selection, crossover, and mutation methods for all three proposed piezocomposite transducers shown in Table 6. Simulations were performed for all cases while considering populations 20, 30, and 40. The characteristics of the transducers for these cases were observed by keeping the front-loading medium water while the back medium was air. The optimized layer thickness and volume fraction obtained by GA variants are then used to find the pressure peak values as a mechanical output of the ODM to calculate the gain bandwidth for all transducers, as shown in Table 7. It is observed that the maximum bandwidth achieved for PZT4d based piezocomposite transducer is 840.979 in the GA5 variant by keeping population 20, for PZT5h based piezocomposite transducer is 1366.647 in the GA4 variant while keeping 20 populations, and for PMN-PT based piezocomposite transducer is 1470.588 in GA2 variant by considering population size 30. The transducer results are also statistically evaluated by taking the mean (avg), standard deviation (std), maximum (max) and minimum (min) fitness values for all GA variants numerically illustrated in Table 8.

### 3.2. Ceramic Based Multilayer Piezocomposite Transducer

In this section, the characteristics of two different multilayer piezocomposite transducers based on active materials are analyzed, simulated, and discussed to improve the sensitivity and gain bandwidth. Two types of piezoelectric materials, PZT4d and PZT5h, were used as active materials along with passive material CY1301. Ten variables in terms of layer thicknesses and volume fractions of these transducers were optimized by GA, and the mechanical output of these optimized piezocomposite transducers was obtained from ODM. The results were achieved in terms of the convergence graph, pressure magnitude, and electrical impedance for populations 20, 30, and 40.

#### 3.2.1. PZT5h-CY1301

The fitness convergence of all cases for the PZT5h-based piezocomposite transducer by various populations 20, 30, and 40 is graphically demonstrated in Figure 4a–c. It is observed that all the transducers based on GA variants show fast convergence for the 40 population. The optimized pressure magnitude peaks for various populations are shown in Figure 4d–f. The transducer shows a higher-pressure peak of 2414 Pascal for population 20, as shown in Table 6.

#### 3.2.2. Piezocomposite of PZT4 d-CY1301

In this section, a piezocomposite five-layer transducer, in which PZT4d is used as an active material, is designed, simulated, and optimized by GA, and then, the mechanical output is evaluated in terms of pressure magnitude and electrical impedance. Simulations were carried out for all GA variants, and the results were obtained for various populations, i.e., 20, 30, and 40. The convergence of fitness function values for all GA variants is graphically shown in Figure 5a–c. The multilayer piezocomposite transducer shows fast fitness convergence for 20 populations and gives the minimum fitness value of 0.001189. Figure 5d–f demonstrates the pressure magnitude for different frequencies. This illustrates higher pressure peaks for the 20 populations shown in Figure 5d.

### 3.3. Piezocomposite PMN-PT-CY1301

In this section, a PMN-PT-CY1301 piezocomposite five-layered transducer is studied for different GA variants to obtain optimized thicknesses and volume fractions. For this purpose, population sizes varying from 20, 30, and 40 were considered, and the results were obtained in terms of pressure magnitude and electrical impedance, graphically shown in Figure 6. The convergence graphs of fitness values of all GA variants are shown in Figure 6a–c. Fast convergence is observed at 30 populations, where a fitness value of 0.0006804 is achieved. The magnitude of output pressure concerning different frequencies is shown in Figure 6d–f, where the higher-pressure peak is observed by considering 30 populations in Figure 6e. The PMN-PT transducer expresses a high-pressure magnitude and lower electrical impedance and gives a maximum bandwidth of up to 2588 mm compared to the other two PZT5h- and PZT4d-based multilayered transducers, as numerically shown in Table 9. Furthermore, the performance among all three transducers based on different materials is studied by Equation (9):(9)$$P\left(\mathrm{dB}\right)=20\times \log_{10}\left(\frac{P}{P_{m}}\right)$$
where *P* is the calculated pressure at each generation and *P_m_* is the achieved maximum pressure value through all generations while *P_m_* for the PZT5h-, PZT4d-, and PMN-PT-based transducers is kept at 2414, 1645, and 2588, respectively. The combined resulting pressures 6 dB are graphically illustrated in Figure 7, where the PMN-PT material-based transducer shows a broad bandwidth of approximately 50 kHz compared to the other transducers.

### 3.4. Result Validation

The comparison of the obtained results by the proposed genetic algorithm optimizer was carried out with the previously proposed stochastic approach simulated annealing (SA) [55]. These comparisons were performed for both PZT5h- and PZT4d-based piezocomposite transducers and are numerically demonstrated in Table 10, where the optimized layer thickness, the ceramic volume fraction of all five layers, and achieved bandwidth are illustrated. The bandwidth obtained by the GA optimizer for the PZT5h-based piezocomposite transducer is 1366 while SA gives a 1130 bandwidth gain; therefore, a 17% improved bandwidth is achieved by the proposed optimizer GA. Similarly, the PZT4d material-based transducer also shows a 5% improved bandwidth; hence, the optimization performed by the proposed GA approach shows a higher bandwidth gain for both piezocomposite transducers based on PZT5h and PZT4d than SA. Furthermore, the newly introduced material PMN-PT-based transducer shows a higher bandwidth than the other PZT5h and PZT4d transducers.

## 4. Conclusions

The proposed evolutionary approach GA has been investigated for sonar applications to improve the piezocomposite transducer performance through ten variable adjustments with center frequencies of 50 kHz. The five nonuniform layer thicknesses and piezoceramic volume fraction for each layer were optimized for PZT5h, PZT4d, and PMN-PT material-based transducers while keeping the water–air loading media. All three transducers are examined through five different cases of GA variants to determine the effectiveness of GA, and a comparison of these transducers is performed with the pre-investigated SA technique. The proposed stochastic GA technique has shown better gain-bandwidth than the SA approach for both PZT5h and PZT4d material-based composite transducers. The novel piezocomposite transducer based on PMN-PT-CY1301 illustrates the highest gain-bandwidth, which is 1469 compared to the other two piezocomposite transducers, and displays an enhanced bandwidth of approximately 50 kHz for −6 dB. Furthermore, the results were validated through a statistical evaluation of fitness function values. Similar transducers can be optimized for various applications, such as biomedical and nondestructive testing. Other evolutionary techniques can also be used according to the fitness function and variability.

## Figures and Tables

**Figure 1 micromachines-14-01677-f001:**
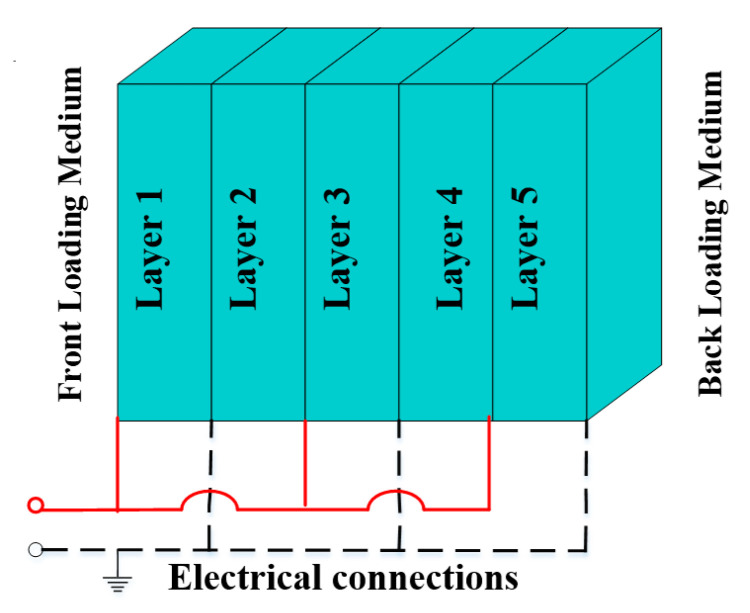
Five layers’ piezocomposite stacked ultrasonic transducer with loading media.

**Figure 2 micromachines-14-01677-f002:**
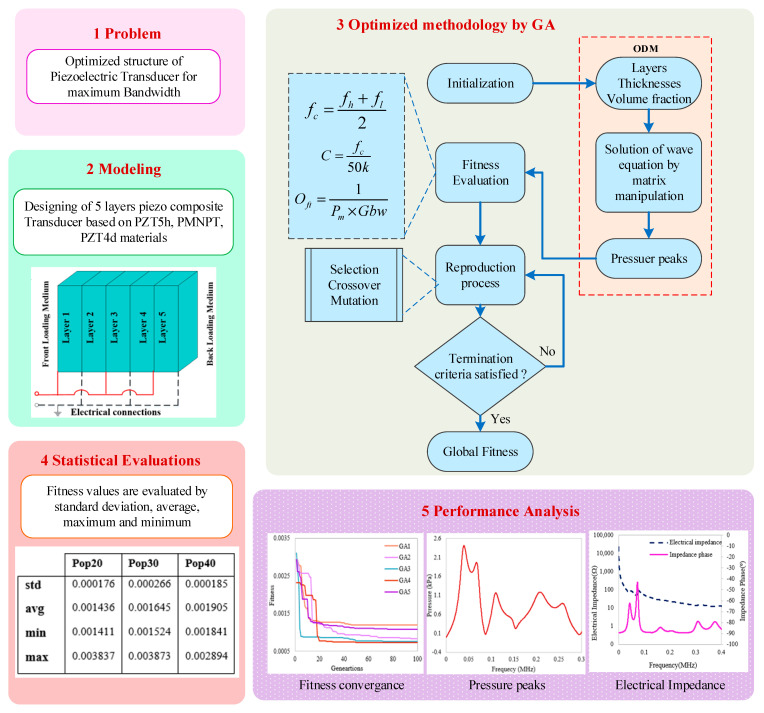
Graphical abstract for the proposed methodology.

**Figure 3 micromachines-14-01677-f003:**
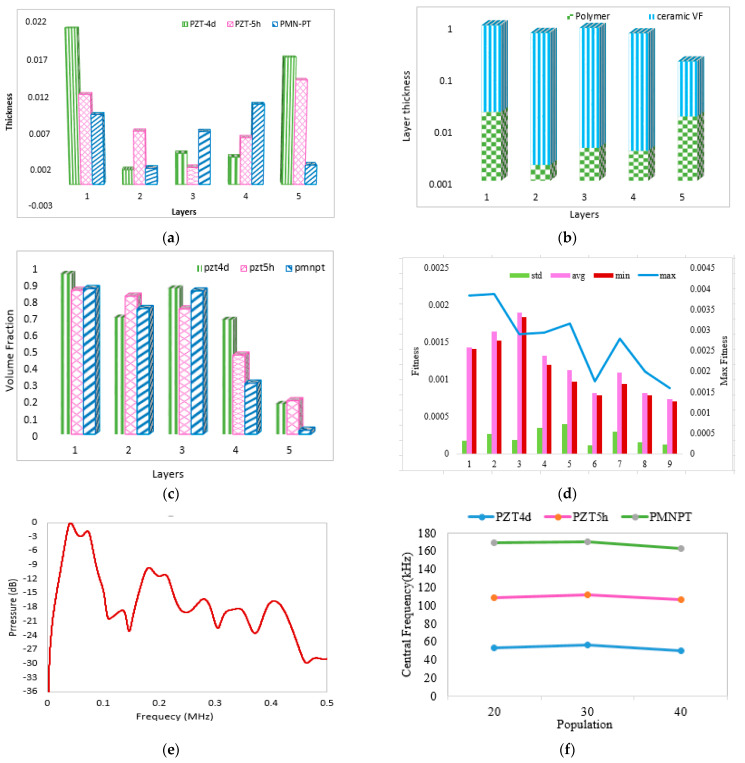
Results comparison for PZT4d, PZT5h, and PMN-PT transducers: (**a**) Layer thicknesses of different materials, (**b**) Ceramic volume fraction in each layer, (**c**) Volume fraction in five layers of different materials, (**d**) Statistical evaluation of fitness values for GA1, (**e**) Peak pressure of PMN-PT in decibel, (**f**) Central frequency for various populations.

**Figure 4 micromachines-14-01677-f004:**
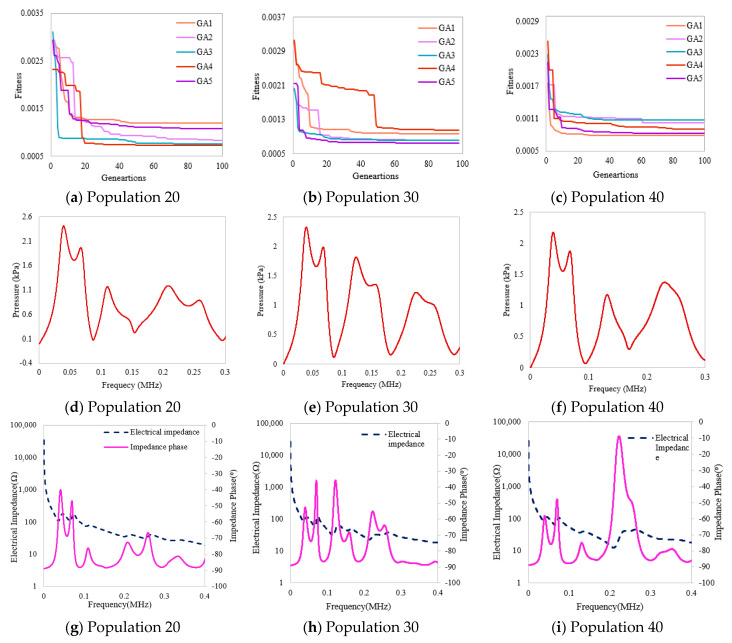
(**a**–**i**) Optimized results of the five-layer PZT5h piezocomposite transducer for various populations.

**Figure 5 micromachines-14-01677-f005:**
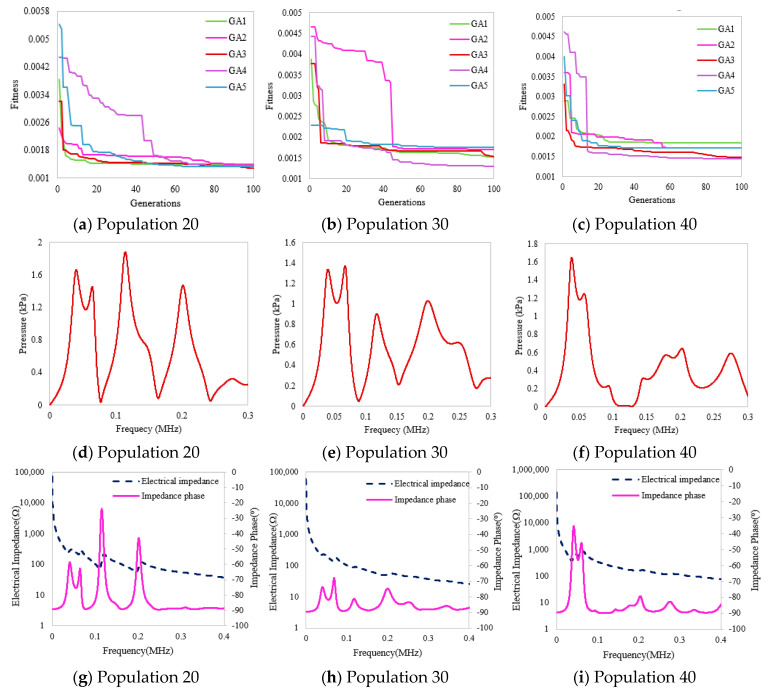
(**a**–**i**) Optimized results of PZT4d based piezocomposite transducer.

**Figure 6 micromachines-14-01677-f006:**
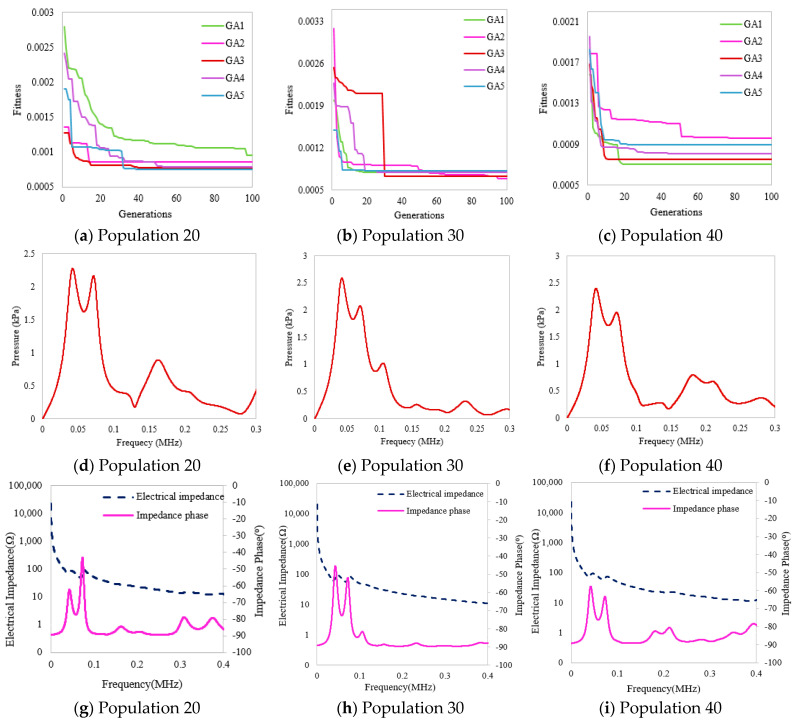
(**a**–**i**) Optimized results of the PMN-PT-based piezocomposite transducer.

**Figure 7 micromachines-14-01677-f007:**
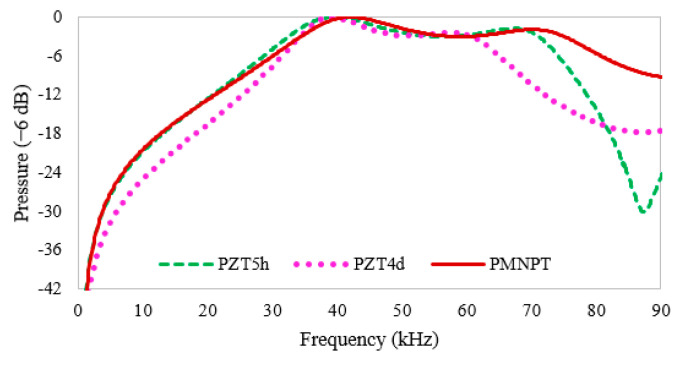
Comparison of output pressure of PZT5h, PZT4d, and PMN-PT-based composite transducers in −6 dB.

**Table 1 micromachines-14-01677-t001:** Characteristics of piezoelectric materials.

Parameters	PZT4d	PZT5h	PMN-PT
Elastic stiffness, $C_{11}^{E}$ (Nm^−2^)	1.369 × 10^11^	1.26 × 10^11^	1.14 × 10^11^
Elastic stiffness, $C_{12}^{E}$ (Nm^−2^)	7.33 × 10^10^	7.95 × 10^10^	1 × 10^11^
Elastic stiffness, $C_{13}^{E}$ (Nm^−2^)	7.218 × 10^10^	8.41 × 10^10^	1.12 × 10^11^
Elastic stiffness, $C_{33}^{E}$ (Nm^−2^)	1.165 × 10^11^	1.17 × 10^11^	1.29 × 10^11^
Piezoelectric stress constant, e^31^ (cm^−1^)	−3.114	−6.5	−3.4
Piezoelectric stress constant, e^33^ (cm^−1^)	14.66	23.3	19.6
Relative permittivity, ${ɛ}_{R}^{T}$	1111	3400	8266
Relative permittivity, ${ɛ}_{R}^{S}$	624	1470	3026
Density, ρ (kg m^−3^)	7568	7500	8050
Material attenuation, α (dB cm^−1^, at 1 MHz)	0.2	0.08	1

**Table 2 micromachines-14-01677-t002:** Material properties of Polymer CY1301.

Material Properties	CY1301
Young’s Modulus (199 Pa)	4.03
Density (kg m^−3^)	1140
Poisson’s Ratio	0.3791
Longitudinal Velocity (m s^−1^)	2565
Relative Permittivity	4
Shear Velocity (m s^−1^)	1132

**Table 3 micromachines-14-01677-t003:** Material properties of front and back loading media.

Material Properties	Water	Air
Acoustic velocity (m/s)	1500	330
Acoustic impedance (MRayl)	1.5	0.000412
Acoustic density (kg/m^3^)	1000	1.25

**Table 4 micromachines-14-01677-t004:** Parameters setting for Genetic Algorithm.

Parameters	Values
Population creation	Constraint dependence
Initial population range	[−5; 5]
Generations	235
Stall generations	50
Scaling fraction	Rank
Ratio	1.2
TolFun	1 × 10^−6^
TolCon	1× 10^−3^
Elite Count	2
Crossover fraction	0.8

**Table 5 micromachines-14-01677-t005:** Optimized results of piezocomposite transducers based on the PZT4d, PZT5h, and PMN-PT materials.

Variables	PZT4d			PZT5h			PMN-PT		
	pop20	pop30	pop40	pop20	pop30	pop40	pop20	pop30	pop40
th1	0.02114	0.01229	0.00945	0.01234	0.00685	0.00651	0.00962	0.00588	0.00502
th2	0.00203	0.00842	0.00897	0.00736	0.01143	0.01003	0.00222	0.00273	0.00958
th3	0.00431	0.00124	0.00584	0.00234	0.00208	0.00197	0.00729	0.00547	0.00218
th4	0.00378	0.00512	0.01154	0.00646	0.00422	0.00331	0.01102	0.00650	0.00727
th5	0.01730	0.01763	0.01508	0.01427	0.01562	0.01819	0.00269	0.00967	0.01074
vf1	0.95311	0.75209	0.86927	0.85598	0.95190	0.78542	0.86544	0.66541	0.28376
vf2	0.69326	0.72781	0.86735	0.82079	0.70958	0.66772	0.74710	0.63463	0.92499
vf3	0.86653	0.91038	0.88213	0.74499	0.57816	0.66771	0.85124	0.77973	0.69970
vf4	0.67993	0.49156	0.72149	0.47006	0.45525	0.75402	0.30368	0.58054	0.66472
vf5	0.18019	0.14869	0.14536	0.19940	0.16337	0.22044	0.02374	0.25488	0.21010
fc	53329.5	56076.1	50430.3	55541.9	55313.1	56228.6	59890.7	58593.7	59280.4
Gbw (Pa)	840.982	775.292	688.778	1366.64	1225.12	1268.27	1338.31	1469.53	1427.15
Fitness	0.00119	0.00129	0.00145	0.00073	0.00081	0.00078	0.00075	0.00068	0.00070

**Table 6 micromachines-14-01677-t006:** Selected GA variants for Simulation.

GA Variants			
Methods	Selection	Crossover	Mutation
GA1	Stochastic Uniform	Heuristic	Adaptive Feasible
GA2	Stochastic Uniform	Arithmetic	Adaptive Feasible
GA3	Remainder	Heuristic	Adaptive Feasible
GA4	Remainder	Arithmetic	Adaptive Feasible
GA5	Stochastic Uniform	Constraint Dependent	Constraint Dependent

**Table 7 micromachines-14-01677-t007:** Gain bandwidth for proposed piezocomposite transducers.

	PZT4d			PZT5h			PMN-PT		
	pop20	pop30	pop40	pop20	pop30	pop40	pop20	pop30	pop40
GA1	708.928	656.340	543.283	837.0736	1027.937	1268.273	1059.883	1266.143	1427.165
GA2	757.915	599.161	583.754	1224.849	1218.28	979.9209	1160.295	1470.588	1040.182
GA3	786.059	675.219	672.400	1303.662	1225.126	930.5275	1293.929	1383.126	1327.915
GA4	735.689	775.29	688.776	1366.647	957.9673	1105.297	1261.416	1262.993	1241.742
GA5	840.979	570.125	582.411	930.0170	1334.133	1216.693	1338.312	1213.592	1114.219

**Table 8 micromachines-14-01677-t008:** Statistical Evaluation of Fitness values.

		PZT4d			PZT5h			PMN-PT		
		Pop20	Pop30	Pop40	pop20	pop30	pop40	pop20	pop30	pop40
GA1	std	0.000176	0.000266	0.000185	0.000351	0.000399	0.000112	0.000297	0.000155	0.000123
	avg	0.001436	0.001645	0.001905	0.001317	0.001128	0.000816	0.001089	0.000820	0.000736
	min	0.001411	0.001524	0.001841	0.001195	0.000973	0.000788	0.000943	0.000790	0.000701
	max	0.003837	0.003873	0.002894	0.002929	0.003160	0.001759	0.002791	0.001994	0.001585
GA2	std	0.000189	0.000986	0.000317	0.000549	0.000279	0.000151	0.000076	0.000227	0.000163
	avg	0.001451	0.002211	0.001853	0.001121	0.000961	0.001113	0.000879	0.000779	0.001046
	min	0.001319	0.001669	0.001713	0.000816	0.000821	0.001020	0.000862	0.000680	0.000961
	max	0.002437	0.004664	0.003608	0.002929	0.002030	0.001725	0.001352	0.003180	0.001786
GA3	std	0.000205	0.000333	0.000188	0.000311	0.000160	0.000150	0.000068	0.000514	0.000124
	avg	0.001362	0.001650	0.001604	0.000856	0.000874	0.001129	0.000791	0.000933	0.000780
	min	0.001272	0.001481	0.001487	0.000767	0.000816	0.001075	0.000773	0.000723	0.000753
	max	0.003209	0.003777	0.003318	0.003109	0.002030	0.002294	0.001272	0.002536	0.001689
GA4	std	0.000811	0.000482	0.000707	0.000489	0.000581	0.000241	0.000256	0.000263	0.000117
	avg	0.001758	0.001484	0.001695	0.000940	0.001611	0.001017	0.000878	0.000867	0.000835
	min	0.001359	0.001290	0.001452	0.000732	0.001044	0.000905	0.000793	0.000792	0.000805
	max	0.004473	0.004433	0.004615	0.002326	0.003160	0.002534	0.002415	0.002279	0.001956
GA5	std	0.000524	0.000142	0.000282	0.000342	0.000240	0.000164	0.000166	0.000087	0.000134
	avg	0.001414	0.001819	0.001795	0.001224	0.000823	0.000889	0.000801	0.000838	0.000929
	min	0.001189	0.001754	0.001717	0.001075	0.000750	0.000822	0.000747	0.000824	0.000897
	max	0.005425	0.002297	0.003996	0.002929	0.002146	0.002142	0.001900	0.001489	0.001833

**Table 9 micromachines-14-01677-t009:** Peak pressure comparison of three transducers.

Material	Fitness Values	Pressure Peaks		
		Pop20	Pop30	Pop40
PZT5h	0.000732	2414.85	2328.98	2177.54
PZT4d	0.001189	1881.48	1371.02	1645.68
PMN-PT	0.000680	2273.92	2588.54	2400.16

**Table 10 micromachines-14-01677-t010:** Comparisons obtained results with SA for both PZT5h- and PZT4d-based piezocomposite transducers.

Optimizer	Number of layers	PZT5h		PZT4d		PMN-PT	
		Thickness	Volume fraction (%)	Thickness	Volume fraction (%)	Thickness	Volume fraction (%)
Genetic Algorithm	1	0.01234	85	0.02114	95	0.00588	66
	2	0.00736	82	0.00203	69	0.00273	63
	3	0.00234	74	0.00431	86	0.00547	77
	4	0.00646	47	0.00378	67	0.00650	58
	5	0.01427	19	0.01730	18	0.00967	25
	Gbw (Pa)	1366		840		1469	
Simulated Annealing	1	0.00831	02	0.01500	5		
	2	0.00929	11	0.00657	44		
	3	0.00845	35	0.00876	99	-	-
	4	0.00875	43	0.00883	77		
	5	0.00954	38	0.01040	87		
	Gbw (Pa)	1130		794			

## Data Availability

No data source is used.

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
