# Peer review of "Neuro-Evolutionary Framework for Design Optimization of Two-Phase Transducer with Genetic Algorithms"

_micromachines, 2023, doi:10.3390/mi14091677_

Round 1
Reviewer 1 Report
1. Figure 1 is the single-most important figure for the paper but currently it is too simple. It requires more labeling. Its caption also requires more explanations and descriptions.
2. Many units were written incorrectly (e.g. Nm-2). Please correct them.
3. The use of subscript was disarrayed too (e.g. e31, e33 in Table 1). Please correct them.
Author Response
Response to Reviewer 1 Comments
Response to the comments on the submitted manuscript ID: micromachines-2512841
Paper Title: Neuro-Evolutionary Framework for Design Optimization of Two-Phase Transducer with Genetic Algorithms
Point 1: Figure 1 is the single-most important figure for the paper but currently it is too simple. It requires more labeling. Its caption also requires more explanations and descriptions.
Response: Agreed with the reviewer comment.Therefore, we have now updated the manuscript by modifying the Figure 1. The caption of figure is also updated and provide more elaboration of given model in the revised manuscript, for better understanding of the readers as sugggested.
Point 2: Many units were written incorrectly (e.g. Nm-2). Please correct them
Response: Thank you so much for identifying such a blunder. The whole paper has now been read thoroughly and removed all the typos and unit ambiguities. Furthermore, for more clarity and to improve the quality of the revised manuscript, the authors have double-checked each unit provided in the manuscript.
Point 3: The use of subscript was disarrayed too (e.g. e31, e33 in Table 1). Please correct them.
Response: Agreed with the statement “The use of subscript was disarrayed too (e.g. e31, e33 in Table 1)” of the worthy anonymous reviewer. Thanks for indicating this ambiguity. Now the standard symbols and units of piezoelectric parameters have been used in Table 1 of the revised manuscript as suggested for better understanding and readability. Please check Table 1 in the revised manuscript.

Reviewer 2 Report
This paper numerically models a five-layer piezocomposite transducer. The goal is to maximize the bandwidth around central frequency 50-60 kHz by varying the layer thickness and the volume fraction of the piezoelectric material in the layer. The results are obtained by using the genetic algorithm and compared with those obtained by stochastic approach simulated annealing method.
I think the paper can be published after major revision.
1) The author should provide a brief description of the genetic algorithm and stochastic approach simulated annealing algorithm. They should explain the terms used in p. 7, such as “population”, “chromosomes”, “recombination of chromosomes”, “the crossover of parents and mutation within chromosomes”, “the genes of individual chromosomes to produce new offspring”, etc. In addition, references have to be given to publications where readers can get comprehensive information about the above algorithms.
2) Why do the authors choose to optimize a 5-layer transducer rather than, e.g., 4- or 6-layer one?
3) How do the authors calculate the values of material constants of layers as functions of volume fraction?
4) Check Table 1. There is a lot of typos in values of material constants and dimension units.
5) Figure 1 and part “Modeling” in Fig. 2: Wires are usually shown by solid lines. Use the standard way to depict the fact that the two wires which cross in a scheme are not in electric contact. Is the voltage not applied to the left-most layer? Check the intersection of wires at the right-most part of the figure.
6) Lines 344-345: “In this section, a single crystal PMN-PT piezocomposite five-layered transducer is 344 studied for different GA variants to obtain optimized thicknesses and volume fractions.” What do the authors mean by “volume fraction” in the case of single crystal layers? A single crystal PMN-PT layer contains only PMN-PT.
7) Check the List of References. There is a lot of incomplete references, e.g., 3,6,73,86,87.
Author Response
Response to Reviewer 2 Comments
Response to the comments on the submitted manuscript ID: micromachines-2512841
Paper Title: Neuro-Evolutionary Framework for Design Optimization of Two-Phase Transducer with Genetic Algorithms
Reviewer 2 General Comments: This paper numerically models a five-layer piezocomposite transducer. The goal is to maximize the bandwidth around central frequency 50-60 kHz by varying the layer thickness and the volume fraction of the piezoelectric material in the layer. The results are obtained by using the genetic algorithm and compared with those obtained by stochastic approach simulated annealing method.
Response: The authors would like to thank the anonymous reviewer for his/her valuable, detailed, and constructive comments that allow us to further improve the quality of our submitted manuscript. All the comments and suggestions are carefully addressed in this document and the paper is accordingly revised.
Point 1: The author should provide a brief description of the genetic algorithm and stochastic approach simulated annealing algorithm. They should explain the terms used in p. 7, such as “population”, “chromosomes”, “recombination of chromosomes”, “the crossover of parents and mutation within chromosomes”, “the genes of individual chromosomes to produce new offspring”, etc. In addition, references have to be given to publications where readers can get comprehensive information about the above algorithms.
Response: Thank you very much to the author for this suggestion. A genetic algorithm and simulated annealing are computational models of evolutionary stochastic algorithms and are well-defined in many books. The reference to these books is now been provided in the reference section of the paper. Furthermore, the explanation of suggested terms is now been added in the revised manuscript.
Point 2: Why do the authors choose to optimize a 5-layer transducer rather than, e.g., 4- or 6-layer one?
Response: The authors would case that just a 5-layer transducer could be optimized. We have carried out the foundation control program, which could easily be modified for 4-, 6- or any number of layers.
Point 3: How do the authors calculate the values of material constants of layers as functions of volume fraction?
Response: The authors used the one-dimensional model (ODM) to calculate the values of material constants as a function of volume fraction.
Point 4: Check Table 1. There is a lot of typos in values of material constants and dimension units.
Response: Thank you so much for identifying such a blunder. The whole paper has now been read thoroughly and removed all the types and unit ambiguities. Furthermore, for more clarity and to improve the quality of the revised manuscript, the authors have double-checked each unit provided in the manuscript. Now the standard symbols and units of piezoelectric parameters have been used in Table 1 of the revised manuscript as suggested for better understanding and readability. Please check Table 1 in the revised manuscript.
Point 5: Figure 1 and part “Modeling” in Fig. 2: Wires are usually shown by solid lines. Use the standard way to depict the fact that the two wires which cross in a scheme are not in electric contact. Is the voltage not applied to the left-most layer? Check the intersection of wires at the right-most part of the figure.
Response: Agreed with the reviewer’s comment. Figure 1 has now been updated in the revised manuscript as suggested. The electrical connections are provided to all layers in a standard way so that the wires cannot interconnect with each other while crossing.
Point 6: Lines 344-345: “In this section, a single crystal PMN-PT piezocomposite five-layered transducer is 344 studied for different GA variants to obtain optimized thicknesses and volume fractions.” What do the authors mean by “volume fraction” in the case of single crystal layers? A single crystal PMN-PT layer contains only PMN-PT.
Response: Thank you to the reviewer for indicating this slip-up. Correction for the pointed sentence has been done in the revised manuscript for a better understanding of the readers. That is: “In this section, a PMN-PT-CY1301 piezocomposite five-layered transducer is studied for different GA variants to obtain optimized thicknesses and volume fractions.”
Point 7: Check the List of References. There is a lot of incomplete references, e.g., 3,6,73,86,87.
Response: Agreed. We have updated the reference section of the revised manuscript by providing citations of original studies in standard format with full details in the body of the draft for better understanding of the readers. The older and less relevant references have now been removed and only recent and relevant references are added in the revised manuscript. Please check the references section of the revised manuscript.

Reviewer 3 Report
The authors of this paper addressed two-Phase transducer with genetic algorithms. The topic seems interesting and fall into the main research area of journal. However, there are three issues to be mentioned.
1) This is not review or survey paper. There are more tha 80 references in this version. So, it is required to reduce the references by excluding some references pulished early and not related closely.
2) Figure should be re-drawn for clearity. Current version does not look professional at all.
3) English should be improved by correcting typoes and grammer erros
English should be improved by correcting typoes and grammer erros
Author Response
Response to Reviewer 3 Comments
Response to the comments on the submitted manuscript ID: micromachines-2512841
Paper Title: Neuro-Evolutionary Framework for Design Optimization of Two-Phase Transducer with Genetic Algorithms
Reviewer 3 General Comment:
The authors of this paper addressed two-Phase transducer with genetic algorithms. The topic seems interesting and fall into the main research area of journal. However, there are three issues to be mentioned.
Response: The authors would like to thank the anonymous reviewer for his/her valuable, detailed, and constructive comments that allow us to further improve the quality of our submitted manuscript. All the comments and suggestions are carefully addressed in this document and the paper is accordingly revised.
Point 1: This is not review or survey paper. There are more than 80 references in this version. So, it is required to reduce the references by excluding some references pulished early and not related closely.
Response: Agreed with the reviewer’s comment that more than 80 references were provided in the manuscript. Therefore, we have updated the reference section of the revised manuscript by providing citations of original studies in the body of the draft for better understanding. The older and less relevant references have now been removed according to the reviewer’s suggestion, and only recent and relevent references are added in the revised manuscript.
Point 2: Figure should be re-drawn for clearity. Current version does not look professional at all.
Response: Thank you very much for this valuable suggestion. The figure in the revised manuscript is now been modified in order to enhance their quality and more readability for the readers.
Point 3: English should be improved by correcting typoes and grammer erros.
Response: Agreed. We have updated the manuscript with a careful, critical, and exhaustive review of the whole draft to improve the linguistic quality by avoiding grammatical/topographical errors, ambiguous sentences, punctuation, and typing mistakes for better understanding of the readers as suggested by the worthy anonymous reviewer. Moreover, for the English/Language corrections, we have requested the most respected Professor Dr. Dumitru Baleanu, Cankaya University, Turkey, and the Institute of Space Sciences, Bucharest, Romania. We appreciate his help, efforts, and support for the improvement of the revised manuscript.

Round 2
Reviewer 2 Report
The authors have made all the necessary corrections, so the paper can be accepted for publications.